# Fragmentation Characteristics of Bubbles in a Throttling Hole Pipe

**DOI:** 10.3390/mi15081025

**Published:** 2024-08-11

**Authors:** Yufeng Zhang, Zhijie Huang, Lixia Sun

**Affiliations:** Mechanical Engineering College, Beihua University, Jilin 132022, China; 15662111552@163.com (Y.Z.); 18370154526@163.com (Z.H.)

**Keywords:** bubble generator, fragmentation, gas-liquid two-phase flow, numerical simulation

## Abstract

To enhance the performance of tubular microbubble generators, the Volume of Fluid (VOF) multiphase flow model in COMSOL Multiphysics was used to simulate the bubble fragmentation characteristics within a throttling hole microbubble generator. The effects of the inlet speed of the throttling hole pipe, the diameter of the throttling hole, and the length of the expansion section on bubble fragmentation performance were analyzed. The results indicated that an increase in the inlet speed of the throttling hole pipe gradually improved the bubble fragmentation performance. However, an increase in the throttling hole diameter significantly reduced the bubble fragmentation performance. Changes in the length of the expansion section had a minor impact on the bubble fragmentation performance. Experimental methods were used to verify the characteristics of bubble fragmentation, and it was found that the simulation and experimental results were consistent. This provides a theoretical basis and practical guidance for the design optimization of tubular microbubble generators.

## 1. Introduction

In recent years, microbubbles have garnered widespread attention due to their unique physical properties, such as small diameter, large specific surface area, stable phase interface, and prolonged residence time in liquids. The size and distribution of bubbles significantly influence the heat and mass transfer processes between gas and liquid phases. Over the past few decades, several typical microbubble generators based on fluid dynamics have been developed, including Venturi tube type [1], swirl type [2], ultrasonic/acoustic pressure type [3], jet type [4], and others [4,5,6,7]. Among these, the tubular compact microbubble generator has emerged as a novel method for microbubble generation and has attracted considerable interest from the academic community [8,9]. The Venturi tube mainly relies on the turbulent flow and cavitation shear crushing in the throat and diffusion sections to generate bubbles. Similarly, the throttling hole type microbubble generator relies on the strong turbulence after the throttling hole and the high-speed shear of the liquid flow to fragment the bubbles. Unlike the Venturi tube, the throttling hole tube does not have an inlet for air and lacks a throat tube. Instead, the gas–liquid mixture flows directly through the throttling hole, where the bubbles are sheared and fragmented.

Tubular microbubble generators are widely used in engineering fields such as ship drag reduction [10], wastewater treatment [11], and mineral flotation [12]. This type of microbubble generator offers advantages such as simple structure, strong adaptability, and large gas-carrying capacity. Significant progress has been made in understanding the mechanism of bubble fragmentation within tubular microbubble generators. For example, Wang Tiefeng [13] developed a model for bubble rupture frequency and sub-bubble size distribution, which can predict the size distribution of bubbles in known turbulent environments. Kolmogorov [14] and Hinze [15] proposed theories on bubble rupture in turbulent fields, suggesting that bubble rupture is caused by the interaction of turbulence and eddies. Based on these theoretical models, it has been concluded that four main mechanisms lead to bubble fragmentation in turbulent fields [16,17,18]: (1) collision of turbulent pulsations and vortices; (2) viscous shear; (3) fluid erosion; (4) instability of large bubble interfaces. Many scholars have also studied the phenomenon and mechanisms of bubble fragmentation inside tubular bubble generators using visualization methods. For example, Kawashima et al. [19] captured transient images of bubble rupture at liquid flow velocities of 9.4 m/s and 21.2 m/s, observing wrinkles in bubbles at low flow velocities. They concluded that flow velocity is an important factor affecting bubble fragmentation. Zhao Liang et al. [20] studied the phenomenon of bubble fragmentation inside Venturi tube bubble generators using visualization techniques, they observed the rapid deceleration and subsequent rupture of bubbles in the diffusion section. Additionally, they investigated the fragmentation of individual bubbles in pipelines, finding that bubbles undergo rapid deceleration, deformation, and rupture in the diffusion section [21]. By comparing the forces acting on bubbles inside Venturi tube bubble generators and traditional pipelines, they determined that turbulence intensity is the main factor causing bubble rupture. Based on theoretical and experimental methods, it can be concluded that parameters such as turbulence intensity, liquid flow velocity, and pressure gradient are crucial factors causing bubble fragmentation. However, limitations such as the simplification of parameters in theoretical models, omission of environmental conditions, randomness, contingency, and measurement errors in experimental processes may lead to an inability to comprehensively and accurately analyze the fragmentation characteristics of bubbles.

With the rapid development of computer technology, Computational Fluid Dynamics (CFD) numerical simulation effectively complements some uncontrollable factors in experimental processes or theoretical models. However, there is currently limited CFD research on the characteristics of bubble fragmentation in turbulent fields. Representative studies include those by Professor Chen Jiaqing’s team from the China University of Petroleum [22,23,24], Professor Yan Changqi’s team from Harbin Engineering University [25,26], and the teams led by Fujiwara and Nomura from Tsukuba University in Japan [16,27]. These research groups have used ANSYS Fluent to numerically simulate fluid flow inside Venturi tubes. 

In light of this, this paper focuses on the throttling hole type microbubble generator, utilizing the simulation software COMSOL 6.2 Multiphysics to establish a computational model. The study employs the level set method for two-phase flow to simulate the throttling hole type microbubble generator, analyzing the influence of different throttling hole diameters, expansion section lengths, and inlet velocities on bubble fragmentation performance. Furthermore, the study compares the size of bubbles generated in simulation and experimental results to validate the bubble fragmentation characteristics.

## 2. Experiment

### 2.1. Experiment Preparation

The throttling hole type microbubble generator processing equipment used in this study is the FDM 3D printer (P1S Combo, Shenzhen Tuo Zhu Technology Co., Ltd., Shenzhen, China). The materials utilized in the research include PVA (water-soluble printing consumables, Shenzhen Tuo Zhu Technology Co., Ltd.), ABS (printing consumables, Shenzhen Tuo Zhu Technology Co., Ltd.), printer bed adhesive (0.5 kg, Shenzhen Tuo Zhu Technology Co., Ltd.), sandpaper (400\1200\2000 grit, Beijing Dongxin Abrasive Tools Co., Ltd., Beijing, China), etc.

When preparing the throttling hole type microbubble generator, PLA material was primarily chosen, and the parts were printed using 3D printing technology, as shown in Figure 1. The manufacturing steps for the throttling hole type microbubble generator are as follows: design the three-dimensional model of the throttling hole type microbubble generator using SolidWorks software; import the model into Magics software to adjust the printing parameters; Calibrate the 3D printer; Apply adhesive on the heated bed of the printer; Print the throttling hole type microbubble generator; Remove the part from the heated bed; Dissolve the print supports in water; use sandpaper to polish the part surface; Place the part in an oven at 45 °C for drying; Obtain a throttling hole type microbubble generator.

### 2.2. Experiment Setup

The throttling hole type microbubble generation device was used as the carrier to construct a complete testing system for observing bubble disintegration, as shown in Figure 2. The bubble disintegration testing system consists of an air compressor (DQE1320-30L, Dongcheng Industrial Group Co., Ltd., Nantong, China), a gas-regulating valve (YQD-4, Shanghai Hugong Valve Factory (Group) Co., Ltd., Shanghai, China), a gas flow meter (DN65 plug-in type, Shanghai Youbang Co., Ltd., Shanghai, China), a water supply tank (140 L), a reactor (28 L acrylic water tank, self-made), a water pump (JTP-4800 32 W, Shanghai Oriental Pump Industry Group Co., Ltd., Shanghai, China), a globe valve (DN25, Shanghai Hugong Valve Factory (Group) Co., Ltd.), a liquid flow meter (DN15A, Changzhou Chengfeng Flowmeter Co., Ltd., Changzhou, China), a high-speed camera (JHSM300f, Canon Group, Tokyo, Japan), LED lights (30 W), and a computer. The air compressor, gas-regulating valve, and gas flow meter form the gas supply unit, which provides stable and adjustable gas pressure to the microbubble generator. The water pump, globe valve, and liquid flow meter form the liquid supply unit, ensuring stable and adjustable liquid flow into the microbubble generator. The high-speed camera, LED lights, and computer constitute the image capture unit for capturing images of bubbles in the reactor.

Compressed air from the air compressor is used as the gas source, while tap water filtered multiple times serves as the experimental water source. A self-made throttle orifice tube, manufactured through 3D printing, is employed to generate fine bubbles in the experimental setup, as shown in Figure 3. The reactor is an organic glass container with dimensions of 25 cm in length, 25 cm in width, and 50 cm in height, providing an effective volume of 28 L. The fine bubbles are captured using a high-speed camera (JHSM300f, Canon Group, Japan) and analyzed using its corresponding software (HotShot SC link).

### 2.3. Image Recognition and Processing

The size of bubble generation is a critical parameter. To investigate the size of bubbles after fragmentation in the throttle hole pipe, MATLAB R2023b image recognition methods are used to identify the number of bubbles, which in turn reflects the size distribution of the bubbles after fragmentation. The method for identifying the number of bubbles is illustrated in Figure 4, where the image processing involves capturing the complete area of bubble fragmentation and trimming off any excess regions. Figure 4a depicts the flowchart of the image recognition process in MATLAB, while Figure 4b shows the steps for identifying the number of bubbles. By following these steps, the number of bubbles can be computed, allowing for the calculation of the average bubble diameter based on the initial area of the bubbles in the two-dimensional cross-section.

## 3. Materials and Methods

### 3.1. Model Structure and Fragmentation Principle

The traditional orifice plate structure, as shown in Figure 5a, is primarily composed of a pipeline and an orifice. Based on the fluid flow pattern within the orifice plate, a new type of orifice plate has been designed with the key dimensional parameters depicted in Figure 5b. The total length of the new orifice plate is set to 220 mm. The length of the outlet section (L) (L = 220 – 80 – D) varies with the length of the expansion section. The inlet section has a diameter of 18 mm and a length of 40 mm, which is followed by a contraction section of 40 mm, an expansion section with a length (D), and an outlet section with a diameter of 22 mm and a length (L). The structure mainly consists of the contraction section, orifice, expansion section, inlet section, and outlet section.

In general, bubbles initially break apart because the external forces acting on them are stronger than the stabilizing forces inside the bubbles. The size of the resulting microbubbles is mainly determined by turbulent inertial forces and surface tension, as described by Equations (1) and (2). Turbulent inertial force is linked to the energy dissipation per unit volume. The condition that the average force exerted on the liquid in turbulence equals the force applied to the same-sized fluid element results in the expression of microbubble size in relation to the Weber number, as indicated in Equation (3).
(1)$$F_{i}\approx \frac{\rho d^{2}}{g}{\left(\frac{\varepsilon dg}{\rho }\right)}^{\frac{2}{3}}$$
(2)$$F_{S}\approx \sigma d$$
(3)$$d\approx {\left(\frac{g\sigma }{\rho }\right)}^{\frac{3}{5}}{\left(\frac{\rho }{\varepsilon g}\right)}^{\frac{2}{5}}$$

In the formula, ρ represents the fluid density (kg/m^3^), d represents the bubble diameter (mm), g represents the positive proportional constant, σ represents the surface tension coefficient, ε represents the energy dissipating rate.

Compared to straight pipe flow, the unique structural form of “contraction + expansion” in orifice plate structures subjects the dispersed phase bubbles to both turbulence-induced forces and other significant forces like pressure gradient forces. Experimental research by Fujiwara et al. from Tsukuba University in Japan [16,27] revealed that bubbles deform at the throat of the Venturi tube. Upon entering the expansion section, the pressure difference at the head and tail of the bubble causes it to break apart. This indicates that static pressure recovery in the expansion section is the primary cause of bubble fragmentation. Preliminary studies by Professor Sun Licheng’s research group at Sichuan University [28], based on the force acting on a single bubble in the Venturi tube, suggest that the sharp increase in pressure gradient forces and additional mass forces in the expansion section lead to a decrease in bubble velocity and intense interaction with the surrounding fluid, thereby promoting bubble deformation and fragmentation.

### 3.2. Governing Equation 

The turbulence model exhibits characteristics of multiple flow scales: the largest occurring length scales, which are dependent on the geometry, the smallest scales of rapid fluctuations, and all scales in between. Using the level set function to track the interface position can more accurately predict the outcomes of jet breakup and topological changes. Therefore, for the study of bubble fragmentation, a Reynolds-Averaged Navier–Stokes (RANS) turbulence model with a level set interface for multiphase flow was selected.
(4)$$\rho \frac{\partial u}{\partial t}+\rho (u\cdot \nabla )u=\nabla \cdot [-pI+\mu (\nabla u+{\left(\nabla u\right)}^{T})+\sigma \kappa \delta n]$$
(5)u·∇=0
(6)$$\frac{\partial \varphi }{\partial t}+u\cdot \nabla \varphi =\gamma \nabla \cdot [\varepsilon \nabla \varphi -\varphi (1-\varphi )\frac{\nabla \varphi }{\left|\nabla \varphi \right|}]$$

In the formula, ρ represents the fluid density (kg/m^3^), u represents the fluid velocity vector (m/s), t represents time (s), p represents the total pressure (pa), I represents the unit matrix, T represents the absolute temperature (K), μ represents the hydrodynamic viscosity (N·s/m^2^), σ represents the surface tension coefficient, κ represents the curvature of the fluid interface, δ represents the two-phase flow interface function, φ represents the horizontal set function, γ represents the re-initialize the parameter, and ε represents the interface thickness control parameters.

The velocity fields calculated from Equations (4) and (5) are substituted into the level set Equation (6) to facilitate the penetration of the continuous phase through the two-phase interface into the dispersed phase. This ensures a smoother transition between the two phases, reducing numerical oscillations in the solutions of the Navier–Stokes equations.

The SST (Shear Stress Transport) model is an improvement over the k-ω and k-ε models, enhancing computational accuracy and offering greater reliability. It is suitable for scenarios involving rapid strain, moderate swirl, and localized complex shear flows, which are essential for simulating bubble fragmentation within a throttling orifice-type microbubble generator. The governing equations for the RANS (Reynolds-Averaged Navier–Stokes) model are as follows:(7)$$\rho \frac{\partial k}{\partial t}+\rho (u\cdot \nabla )k=\nabla \cdot [(\mu +\frac{\rho c_{\mu }k^{2}}{\varepsilon \sigma_{k}})\nabla k]+\rho c_{\mu }\frac{k^{2}}{\varepsilon }[\nabla u:(\nabla u+{\left(\nabla u\right)}^{T})]-\rho \varepsilon$$
(8)$$\rho \frac{\partial \varepsilon }{\partial t}+\rho (u\cdot \nabla )\varepsilon =\nabla \cdot [(\mu +\frac{\rho c_{\mu }k^{2}}{\varepsilon \sigma_{\varepsilon }})\nabla \varepsilon ]+c_{1}\rho s\varepsilon -c_{\varepsilon 2}\frac{\rho \varepsilon^{2}}{k+\sqrt{v\varepsilon }}$$

In the formula, k represents the turbulent kinetic energy (J), $c_{\mu }$ represents the model constant, ε represents the rate represents turbulence dissipation, $\sigma_{k}$ represents the constant in the kinetic energy model of turbulence, “:” represents the contraction operator, $\sigma_{\varepsilon }$ represents the constant in the turbulent dissipation rate model, and v represents the velocity scalar (m/s).

### 3.3. Numerical Simulation Methods

#### 3.3.1. Geometric Model and Grid Partitioning

The numerical simulation employs a geometric model with structural dimensions identical to those of the experimental throttling orifice-type microbubble generator. Taking the center of the inlet cross-section as the reference zero point, the three-dimensional graphics and grid partitioning are shown in Figure 6. The computational domain grid consists of an O-shaped hexahedral structure with grid density increased in the contraction section, throttling orifice, and expansion section. The computational domain is three-dimensional.

#### 3.3.2. Selection of Numerical Simulation Model and Boundary Condition Settings

To investigate the deformation and fragmentation process of bubbles in the throttling orifice pipe flow, the VOF (Volume of Fluid) model is chosen for the multiphase flow simulation. The turbulence model used in the COMSOL commercial software is based on the Reynolds-averaged equations, with commonly used types including the standard k-ε model, the RNG k-ε model, and the Reynolds stress model [29,30]. The RNG k-ε model is selected in this study, as it adds a condition in the ε equation compared to the standard k-ε model, effectively improving accuracy and accounting for turbulent eddies, thus providing higher credibility and accuracy. The Reynolds stress model, while suitable for complex and strong turbulent conditions due to its detailed turbulence representation, has higher memory consumption and poor convergence due to the strong coupling between equations. Therefore, the RNG k-ε turbulence model is chosen to simulate the turbulent flow process in the throttling orifice pipe. Referring to the numerical simulation settings used by relevant scholars [31,32,33], and considering that this study focuses on the influence of Reynolds number on bubble fragmentation and pressure drop change, the boundary conditions are set as follows: the inlet boundary condition is set as the velocity inlet, and the outlet boundary condition is set as the pressure outlet. Additionally, to simplify the model and eliminate the influence of other irrelevant factors on the simulation results, the walls are set as no-slip walls. Turbulence is defined using the hydraulic diameter and turbulence intensity. The hydraulic diameter is equal to the entrance diameter of the Venturi tube, and the calculation formula for turbulence intensity (I) is given by Equation (9).
(9)I=0.16×Re−0.125

The working medium consists of water (main phase) and air (secondary phase). The density of the main phase water is 998 kg/m^3^, and the density of the secondary phase air is 1.225 kg/m^3^. The interfacial viscosity coefficient between the gas–liquid phases is 0.075 N/m. The control volume method is utilized to discretize the governing equations. The pressure is discretized using the PRESTO! scheme, while the momentum is discretized using a second-order upwind scheme. A single bubble with a diameter of 4 mm is introduced at the inlet section along the axis. This choice is made due to the high computational demands of multi-bubble cavitation simulation, prompting us to focus on a single bubble. The bubble diameter of 4 mm is selected based on experimental observations, where gas-phase flow and liquid-phase flow through the T-junction produced bubbles with diameters ranging from 3 to 5 mm. Notably, bubbles with a 4 mm diameter were the most prevalent, accounting for 60% of the total observed bubbles.

#### 3.3.3. Mesh Independence Verification

The number of grids significantly impacts the accuracy of numerical calculations. Numerical simulation results gain significance only when the grid reaches a certain quantity, and key calculation results show minimal changes with further grid increase. To verify mesh independence, four groups of models with grid numbers 85,427; 114,362; 206,523; and 334,562 were created. As shown in Figure 7, the pressure distribution along the central axis of the throttling orifice pipe is recorded. It is observed from the figure that as the grid number increases to 206,523, the pressure distribution along the axis stabilizes, and further increasing the grid number has a minimal impact on the results. Due to a noticeable deviation in pressure at 150 mm, the turbulence dissipation rate at this cross-section was chosen to further validate mesh independence. The average turbulence dissipation rate and relative deviation for each group of models were calculated and presented in Table 1. The results indicate that as the grid number increases, the relative deviation gradually decreases. When the grid number increases from 114,362 to 206,523, the relative deviation is 0.83%. When the grid number increases from 206,523 to 334,562, the relative deviation is 0.38%. A comparison of the pressure distribution along the central axis and the average turbulence dissipation rate on the cross-sectional plane shows that a grid number of 206,523 is sufficient to meet the requirements of mesh independence.

#### 3.3.4. Physical Model Testing and Verification

To verify the accuracy of the Computational Fluid Dynamics (CFD) numerical simulation model established for physical model testing, an experimental setup was constructed as shown in Figure 8. A comparison and analysis were conducted between the experimental results and the CFD numerical simulation results. The main components of the experimental setup include a water tank, centrifugal pump, throttling orifice pipe, flow meter, pressure gauge, and valve. During the experiment, the water flow rate was adjusted by changing the centrifugal pump speed and the opening of the side valve. The inlet and outlet pressures of the throttling orifice pipe were observed and recorded. To minimize random errors, the experiment was repeated three times, and the average values were taken. The experimental results were compared with the CFD numerical simulation results, as shown in Table 2. From the table, it can be seen that when using the RNG k-ε turbulence model, the maximum deviation between the average pressure drop at the inlet and outlet sections and the experimental values is only 1.15%. Therefore, it is evident that the established CFD numerical simulation model meets the accuracy requirements for calculations.

To further validate the reliability of the CFD numerical simulation results, following the experimental procedure outlined in reference [24], a Reynolds number of Re = 20,000 was set at the inlet. After achieving stability in the single-phase flow field, a 4 mm initial bubble was injected (patched) at a distance of x = 125 mm from the throat entrance, and its movement in the throat section and the breakup segment was observed. The post-processing results of the CFD numerical simulation are shown in Figure 9. According to the graph, the injected bubble maintains its shape in the throat section, but it undergoes significant deformation and fragmentation upon entering the expansion section. In the Venturi channel, the CFD numerical simulation of the bubble fragmentation process is generally consistent with the experimental results, although there are some differences in the extent of bubble breakup in the expansion section compared to the experimental data. Analysis suggests that this is mainly due to the fact that during the experiment, the initial bubbles generated by the injection tube have a certain initial velocity, while the initial bubble injected by the Patch in the CFD numerical simulation has a velocity of zero. Additionally, the current numerical simulation approach is unable to fully replicate the turbulent flow field within the channel, leading to differences in the simulated bubble morphology in the expansion section compared to the observed experimental results [34].

#### 3.3.5. Simulation Model

A geometric model for the numerical simulation of the new throttling orifice tube has been established based on its actual dimensions, as shown in Figure 5b. The total length of the new throttling orifice tube is set at 220 mm, with the length of the outlet section (L) determined as (220 − 80 − D), varying with the length of the expansion section (D). The inlet section has a diameter of 18 mm and a length of 40 mm, which is followed by a 40 mm contraction section, a (D)-length expansion section, and an outlet section with a diameter of 22 mm and a length of (L). In the simulation process, the inlet diameter, outlet diameter, and total length of the throttling orifice tube remain constant, while the orifice diameter, expansion section length, and inlet velocity are adjusted to investigate the bubble rupture characteristics inside the throttling orifice tube. Table 3 below outlines the parameter settings for the model.

## 4. Analysis of Numerical Calculation Results

### 4.1. The Fragmentation Process of Bubbles in the Throttling Orifice Tube

To explore the fragmentation process of bubbles in the throttling orifice tube, the inlet flow velocity was set to 2 m/s, the orifice diameter was set to 4 mm, and the expansion section length was set to 40 mm. The simulation results shown in Figure 10a indicate that bubbles inside the throttling orifice tube undergo three stages: deformation, stretching, and fragmentation. During the numerical simulation, bubbles experience slight deformation in the inlet section, which is known as the deformation zone. As bubbles move from the contraction section to the expansion section, they undergo severe deformation and stretch into the stretching zone. Upon entering the expansion section, the increasing cross-sectional area leads to a growing adverse pressure gradient, disrupting the pressure balance inside and outside the bubbles, causing initial fragmentation. Figure 10b shows that the velocities at the center of the throttling orifice tube and near the wall differ, creating a velocity difference that generates vortices. The presence of vortices intensifies flow field disturbances, leading to increased bubble fragmentation. Vortices can be observed in the outlet section, resulting in varying sizes of fragmented bubbles and uneven distribution.

### 4.2. The Influence of Inlet Velocity on the Characteristics of Bubble Fragmentation

By setting the inlet velocities at 1 m/s, 2 m/s, 3 m/s, and 4 m/s, and selecting bubble fragmentation times of 0.16 s, 0.08 s, 0.056 s, and 0.04 s, respectively, transient diagrams for observing bubble fragmentation were created. The throttling orifice diameter is 4 mm, and the expansion section length is 40 mm. The aim is to explore the effect of different inlet velocities on the characteristics of bubble fragmentation. The results are illustrated in Figure 11. From the figure, it can be observed that as the inlet velocity increases, the fragmentation time of the bubbles decreases, and the bubble size also decreases. When the inlet velocity exceeds 2 m/s, the rate of reduction in bubble size slows down.

Changes in the flow field inside the throttling orifice tube reflect the characteristics of bubble fragmentation with variations in velocity significantly accelerating the time of bubble fragmentation. To ensure that the bubbles fragment at roughly the same position after passing through the throttling orifice tube, times of 0.16 s, 0.08 s, 0.056 s, and 0.04 s were selected to investigate the changes in the flow field inside the throttling orifice tube, as shown in Figure 12. As the inlet velocity of the throttling orifice tube increases, the fluid velocity, pressure drop, turbulent dissipation rate, and turbulent kinetic energy also increase significantly. This indicates that as the inlet velocity increases, the fragmentation performance of bubbles improves. By appropriately increasing the inlet velocity (pressure), the fragmentation performance of bubbles can be further enhanced. Considering the analysis of bubble size generation and energy consumption, an inlet velocity of 2 m/s is chosen as the reference velocity for the throttling orifice tube.

### 4.3. Effect of Throttling Orifice Diameter

In the throttling orifice tube, the throttling ratio (the ratio of orifice diameter to inlet diameter, (r/D)) is an important parameter influencing bubble fragmentation characteristics. To investigate the effect of the throttling ratio on bubble fragmentation characteristics, different orifice diameters of 3 mm, 4 mm, 5 mm, and 6 mm were set with a fragmentation time of 0.07 s to obtain throttling orifice tubes with different throttling ratios (3/18, 4/18, 5/18, 6/18). The fragmentation performance of bubbles under different orifice diameters was explored with an inlet velocity of 2 m/s and an expansion section length of 40 mm. As shown in Figure 13, with an increase in the throttling ratio, the size of generated bubbles initially decreases and then increases. When the throttling ratio exceeds 4/18, bubble fragmentation performance noticeably decreases. An increase in the throttling ratio leads to a reduction in fluid velocity at the orifice, which decreases the pressure difference at both ends of the orifice and weakens the shearing fragmentation effect on bubbles. Therefore, within a certain range, a higher throttling ratio results in poorer bubble fragmentation performance in the throttling orifice tube.

Since the fragmentation process of bubbles occurs in the expansion section and the exit section, numerical values of bubbles passing through the orifice (at 0.06 s) were selected to explore the impact of changes in the internal flow field under different orifice diameters on bubble fragmentation characteristics, as shown in Figure 14. Fluid velocity, pressure, turbulent dissipation rate, and turbulent kinetic energy are the main parameters affecting bubble fragmentation performance, and from the figure, it is evident that these parameters primarily change after the contraction section. As the throttling ratio increases, the fluid velocity, pressure difference, turbulent dissipation rate, and turbulent kinetic energy gradually decrease. Combined with Figure 13, it is evident that a higher throttling ratio leads to poorer bubble fragmentation performance. Conversely, a smaller throttling ratio leads to better bubble fragmentation performance. However, there is a limit to reducing the throttling ratio. When the orifice diameter is smaller than this limit, it exacerbates bubble collisions, leading to coalescence, which decreases the bubble quantity and increases the bubble size. Based on the above considerations, a 4 mm orifice diameter is selected as the reference diameter for the throttling orifice.

### 4.4. Effect of Expansion Length on Bubble Fragmentation Characteristics

Setting the expansion length to 30 mm, 35 mm, 40 mm, and 45 mm, respectively, while keeping other parameters constant, the effects of different expansion lengths on bubble fragmentation performance were investigated. The predicted results are shown in Figure 15.

As the length of the expansion section increases, the size of the generated bubbles slowly increases. It can be observed from the overall range of changes that the effect of varying the length of the expansion section on bubble size is relatively small. From Figure 16, it is evident that the changes in these parameters mainly occur after the contraction section. With the increase in the length of the expansion section, the trends in fluid velocity, pressure, turbulent dissipation rate, and turbulent kinetic energy are generally similar. Combined with Figure 15, 30 mm is chosen as the reference length for the expansion section of the throttling orifice tube.

## 5. Experimental Results Analysis 

During the performance testing experiment of the throttling orifice tube, the gas-liquid ratio is an important parameter that not only affects the bubble generation particle size but also influences the gas content. With an inlet velocity of 2 m/s, a throttle aperture of 4 mm, and an expansion section length of 40 mm, the influence of the gas-liquid ratio on the bubble fragmentation size was investigated. The experimental results, as shown in Figure 17, indicate that bubble diameters are mainly distributed between 50–80 μm. When the gas and liquid phases pass through the throttling orifice, too little gas can reduce the gas content, while an excess of gas can disrupt the pressure balance before and after the throttling orifice, leading to uneven and larger bubble diameters. Therefore, it is crucial to maintain a gas-liquid ratio of 6% throughout the experiment.

### 5.1. Effect of Inlet Velocity on Bubble Fragmentation Characteristics

To investigate the effect of inlet velocity on bubble fragmentation size in the throttling orifice tube, different inlet velocities of 1 m/s, 2 m/s, 3 m/s, and 4 m/s were tested while keeping the other parameters constant. The performance testing of the throttling orifice tube was conducted using the experimental setup, and the average bubble size was obtained, as shown in Figure 18. With the increase in inlet velocity, there is a gradual decrease in average bubble size, and this decrease tends to level off gradually. By comparing and analyzing the changes in bubble size between the experimental results and the simulation results, it is observed that the average deviation between the two sets of data is within 8%. Therefore, it can be concluded that the two are consistent. The fine bubbles generated by the experiment are shown in Figure 19.

### 5.2. Effect of Orifice Diameter on Bubble Fragmentation Characteristics

By taking orifice diameters of 3 mm, 4 mm, 5 mm, and 6 mm, respectively, while keeping other parameters constant, the effect of orifice diameter on bubble fragmentation performance was investigated. Performance testing of the throttling orifice tube was conducted using the experimental setup, and the average bubble size was obtained, as shown in Figure 20. It can be observed from the graph that with an increase in orifice diameter, the size of generated bubbles tends to increase. Between orifice diameters of 3 mm and 4 mm, there is almost no change in the size of generated bubbles. The overall analysis results indicate that the deviation is within 5%, demonstrating that the experimental results are consistent with the simulation results.

### 5.3. Effect of Expansion Segment Length on Bubble Fragmentation Characteristics

To investigate the effect of expansion segment length on bubble fragmentation size in the throttling orifice tube, expansion segment lengths of 30 mm, 35 mm, 40 mm, and 45 mm respectively were considered while keeping other parameters constant. Performance testing of the throttling orifice-type fine bubble generator was conducted using the experimental setup, and the average bubble size was obtained, as shown in Figure 21. With an increase in expansion segment length, the size of generated bubbles slowly increases, and the range of bubble size variation does not exceed 30 μm. By comparing and analyzing the changes in bubble size between the experimental results and the simulation results, it is observed that the average deviation between the two sets of data is within 10%. Therefore, it can be concluded that the two are consistent.

## 6. Conclusions

This article conducted a study on the throttling orifice-type bubble generator using CFD technology, analyzing the influence of variations in inlet velocity, orifice diameter, and expansion segment length on bubble fragmentation performance. The following conclusions were drawn:(1)The internal bubble fragmentation in the throttle orifice tube can be divided into three stages: deformation, stretching, and fragmentation. Changes in fluid velocity, pressure, turbulent dissipation rate, and turbulent kinetic energy mainly occur in the contraction and expansion segments. The difference between wall velocity and central velocity is the main reason for the uneven bubble size and different bubble distribution.(2)As the throttle ratio decreases, the fragmentation performance of bubbles improves, but there is a limit to the decrease in the throttle ratio. In this study, a throttle ratio of 4/18 (orifice diameter of 4 mm) was chosen as the optimal throttle ratio.(3)The variation in expansion segment length of the throttle orifice tube has a relatively small effect on bubble fragmentation performance. In this study, an expansion segment length of 30 mm was selected for the throttle orifice tube.(4)With an increase in the inlet velocity of the throttle orifice tube, the fragmentation performance of bubbles also improves. In this study, an inlet velocity of 2 m/s was chosen for the throttle orifice tube.

## Figures and Tables

**Figure 1 micromachines-15-01025-f001:**
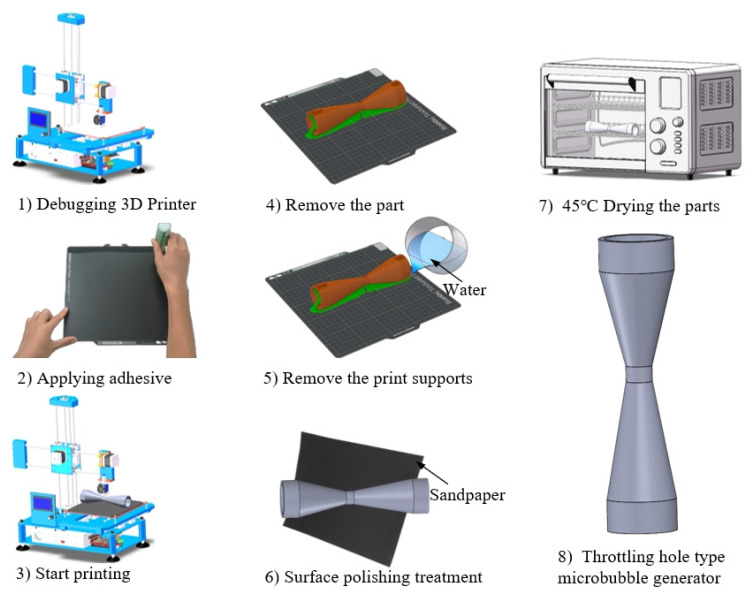
The fabrication process of throttling hole type microbubble generator.

**Figure 2 micromachines-15-01025-f002:**
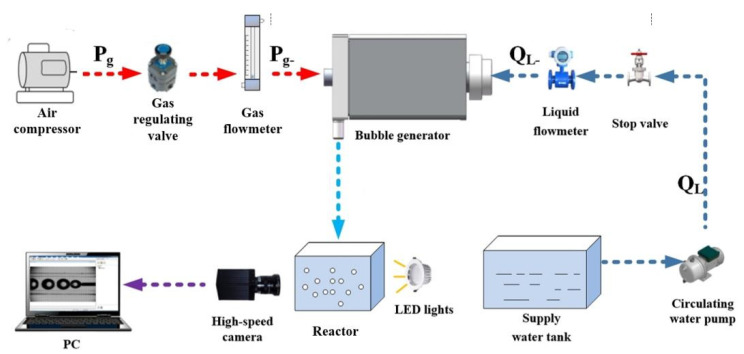
Bubble fragmentation experimental principle diagram.

**Figure 3 micromachines-15-01025-f003:**
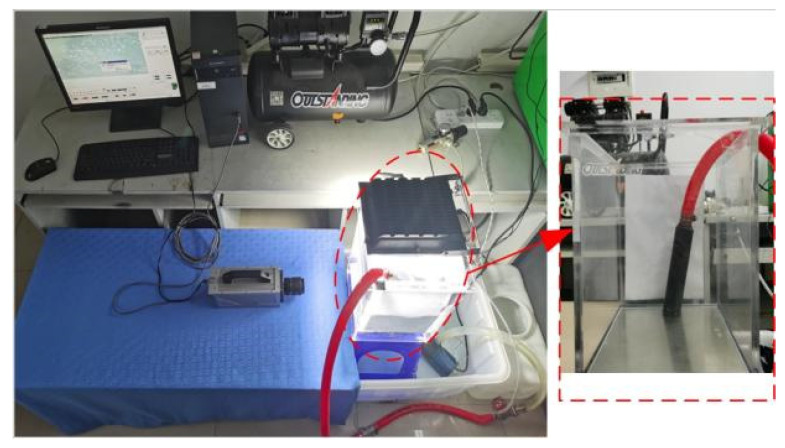
Experimental device diagram.

**Figure 4 micromachines-15-01025-f004:**
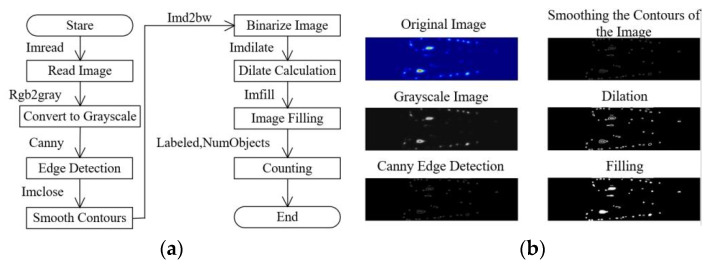
Calculation method for the number of bubbles inside the orifice type. (**a**) Quantity recognition flowchart; (**b**) Quantity recognition steps.

**Figure 5 micromachines-15-01025-f005:**
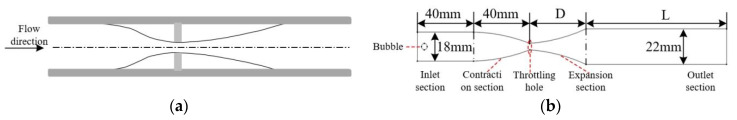
Orifice plate structure diagram: (**a**) Traditional throttling hole pipe structure diagram; (**b**) New type throttling hole pipe structure diagram.

**Figure 6 micromachines-15-01025-f006:**
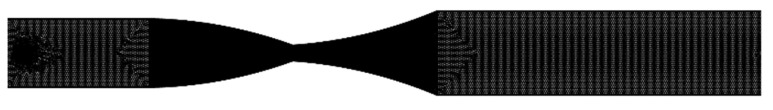
The grid partitioning of the model.

**Figure 7 micromachines-15-01025-f007:**
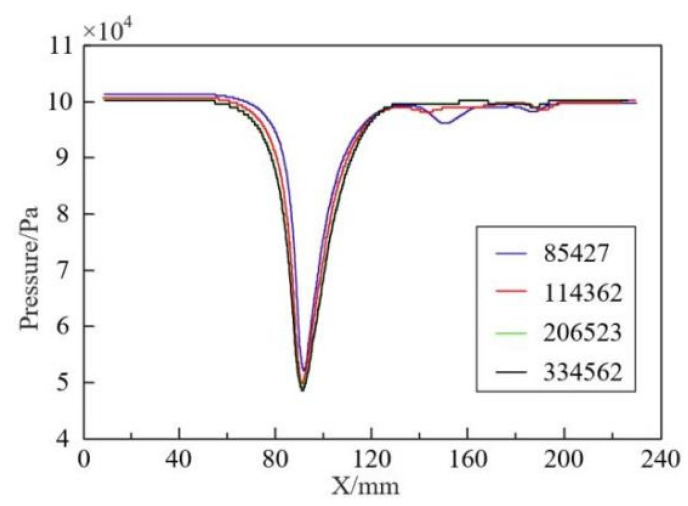
Grid independence verification.

**Figure 8 micromachines-15-01025-f008:**
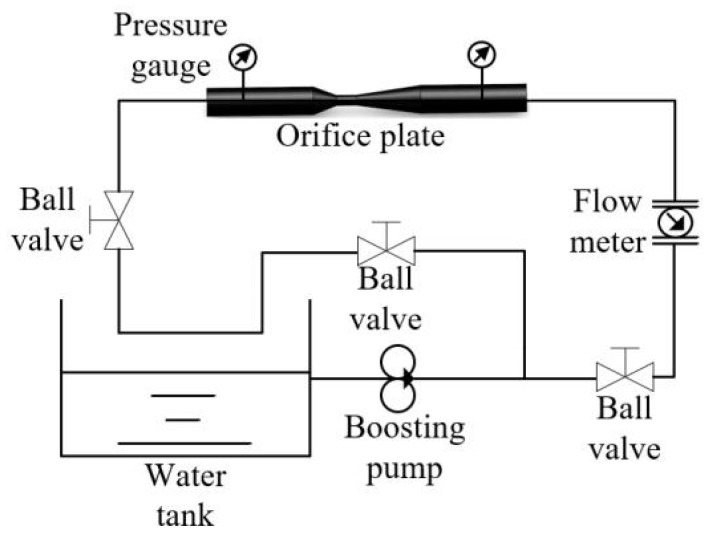
Schematic of the Experimental Procedure.

**Figure 9 micromachines-15-01025-f009:**
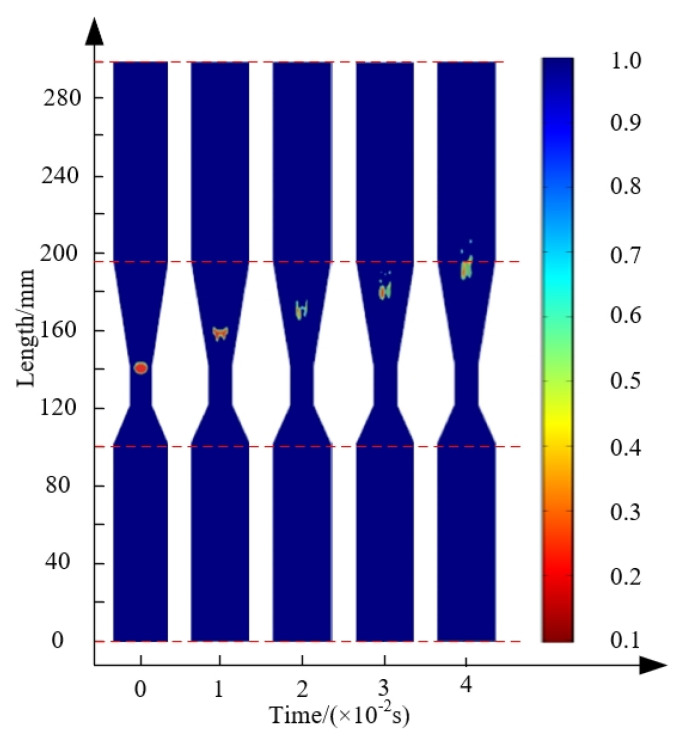
Cloud diagram of the bubble fragmentation process.

**Figure 10 micromachines-15-01025-f010:**
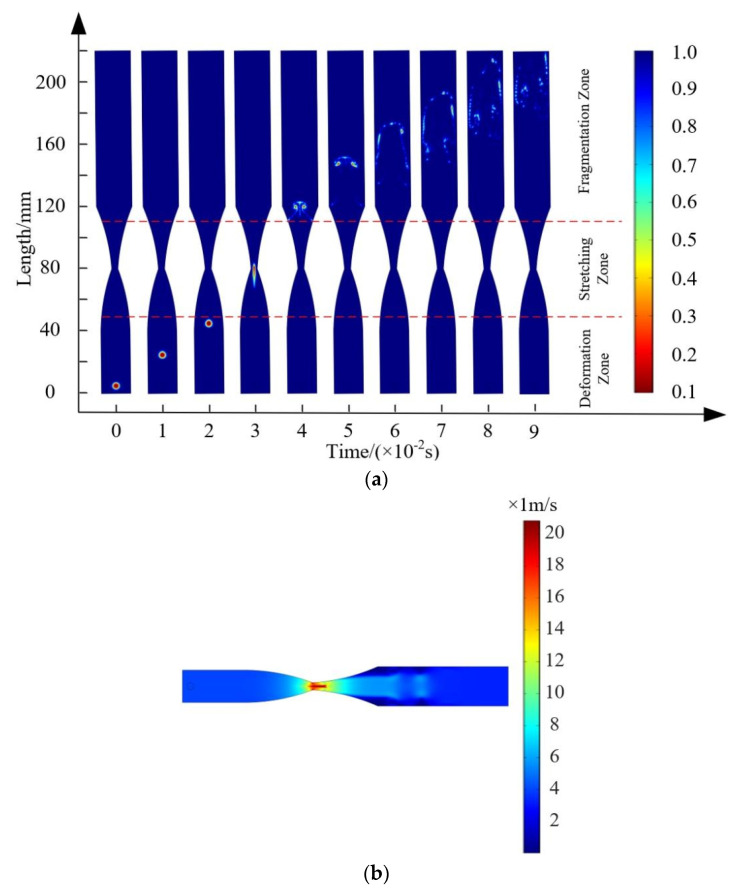
Fragmentation transients and velocity clouds of bubbles inside a throttle orifice tube: (**a**) transitional diagram of bubble development in the throttling orifice tube; (**b**) velocity field map of the flow inside the throttling orifice tube.

**Figure 11 micromachines-15-01025-f011:**
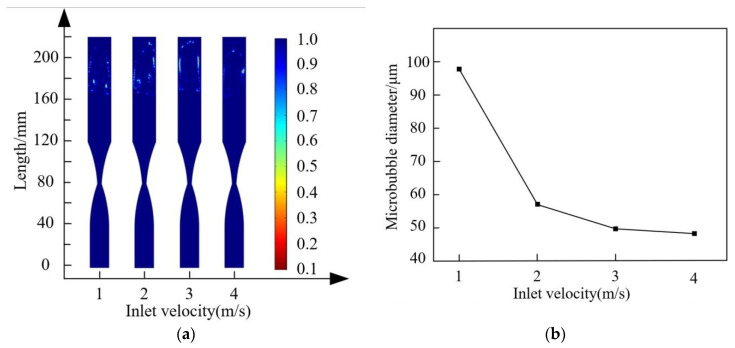
Fragmentation characteristics of bubbles at different inlet velocities: (**a**) transient diagrams of bubble fragmentation under different inlet velocities; (**b**) diameter diagram of bubbles after fragmentation at different inlet velocities.

**Figure 12 micromachines-15-01025-f012:**
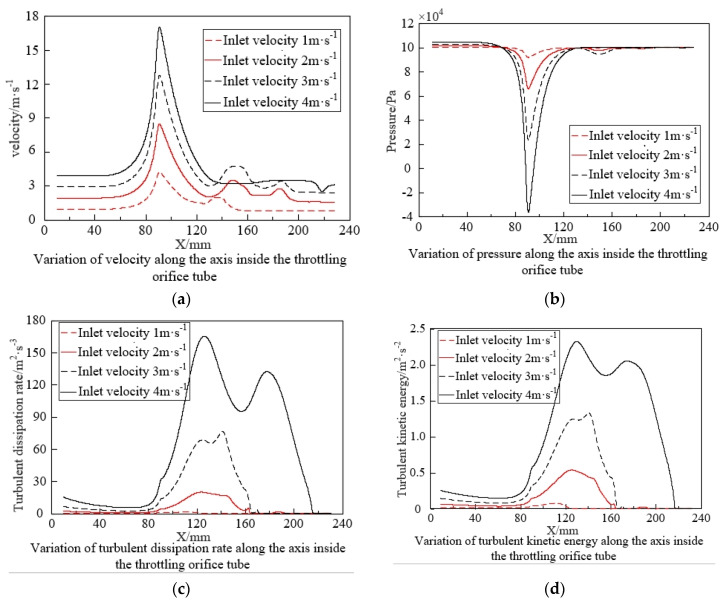
Fragmentation characteristics of bubbles at different inlet velocities.

**Figure 13 micromachines-15-01025-f013:**
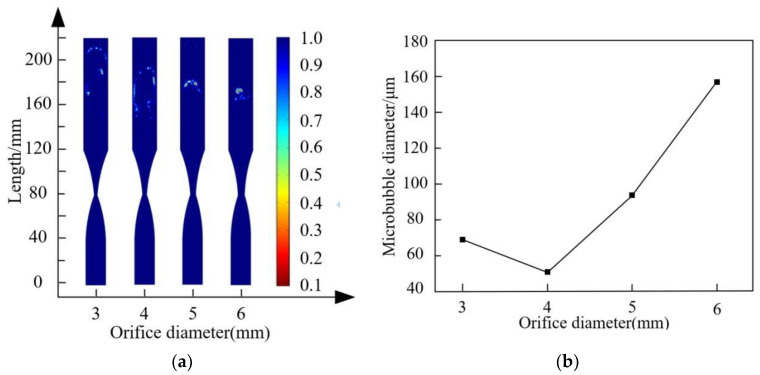
Bubble fragmentation performance under different throttling aperture: (**a**) Transient diagram of bubble fragmentation under different orifice diameters; (**b**) Diameter diagram of bubbles after fragmentation under different orifice diameters.

**Figure 14 micromachines-15-01025-f014:**
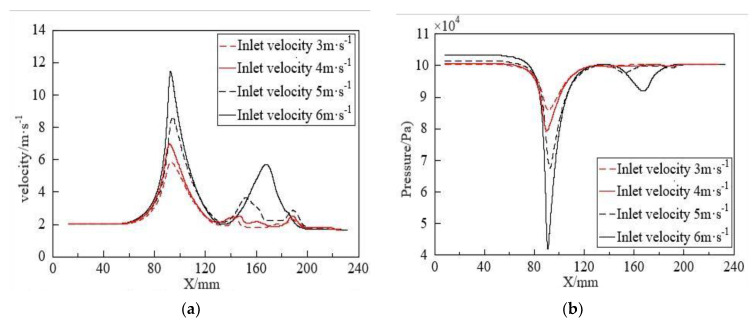
Flow characteristics of fluid under different throttling apertures: (**a**) Variation in velocity along the axis inside the throttling orifice tube; (**b**) Variation in pressure along the axis inside the throttling orifice tube; (**c**) Variation of turbulent dissipation rate along the axis inside the throttling orifice tube; (**d**) Variation of turbulent kinetic energy along the axis inside the throttling orifice tube.

**Figure 15 micromachines-15-01025-f015:**
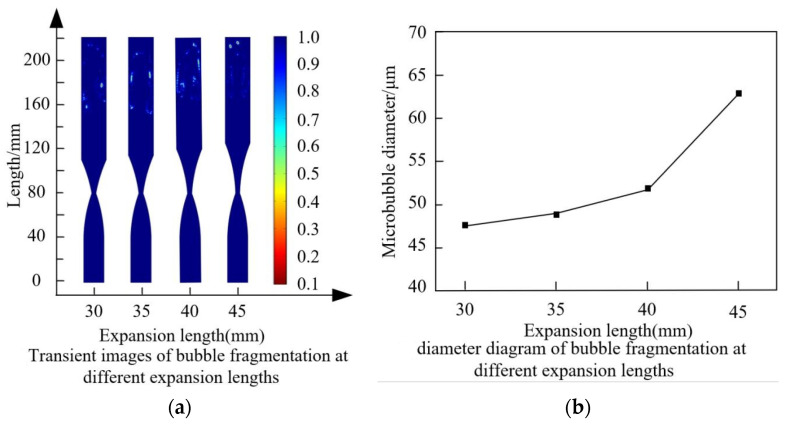
Fragmentation characteristics of bubbles under different expansion section lengths.

**Figure 16 micromachines-15-01025-f016:**
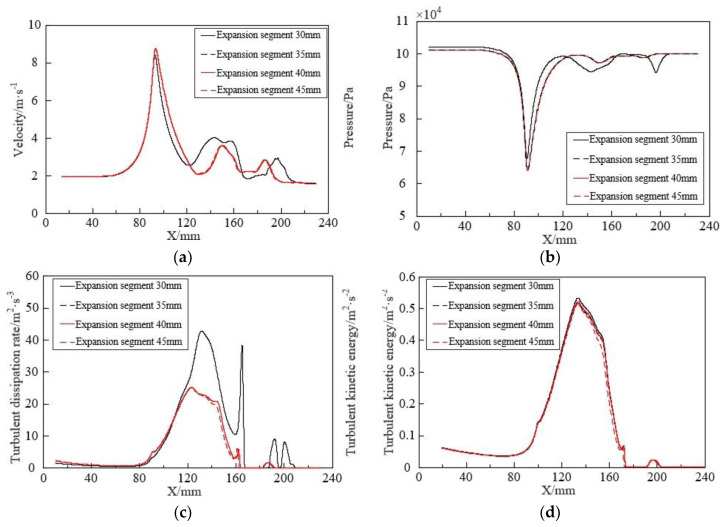
Flow characteristics of fluid under different expansion section lengths: (**a**) Variation of velocity along the axis inside the throttling orifice tube; (**b**) Variation of pressure along the axis inside the throttling orifice tube; (**c**) Variation of turbulent dissipation rate along the axis inside the throttling orifice tube; (**d**) Variation of turbulent kinetic energy along the axis inside the throttling orifice tube.

**Figure 17 micromachines-15-01025-f017:**
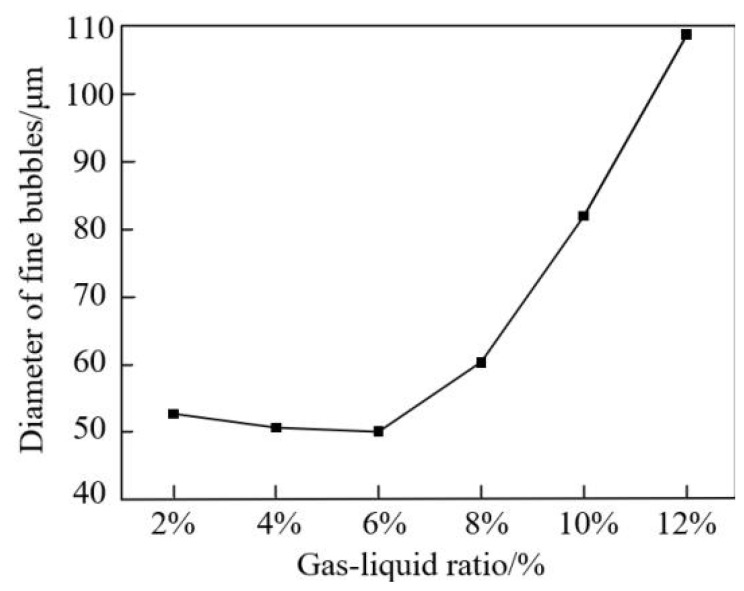
Effect of gas-liquid ratio on bubble diameter in orifice tube.

**Figure 18 micromachines-15-01025-f018:**
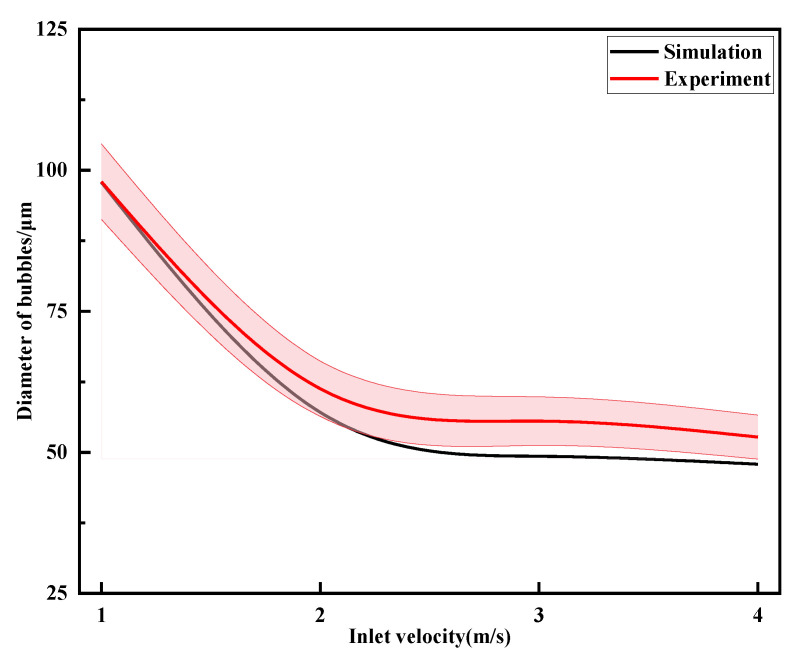
Results of changes in bubble particle size under different inlet velocities.

**Figure 19 micromachines-15-01025-f019:**
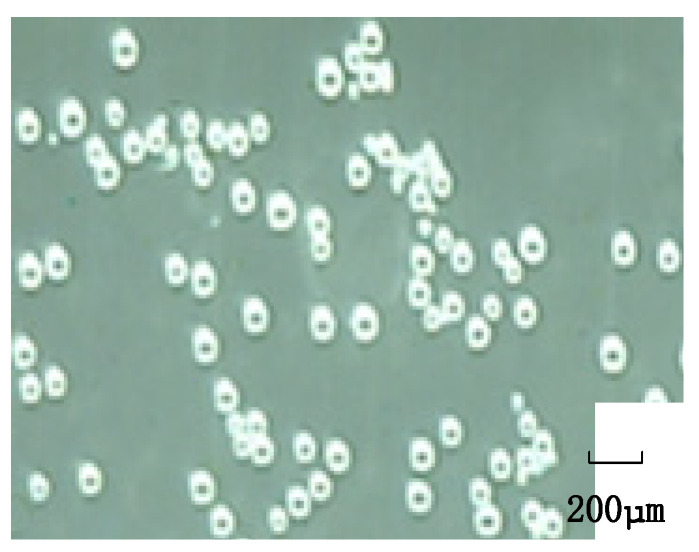
The bubble generator produces microfine air bubbles.

**Figure 20 micromachines-15-01025-f020:**
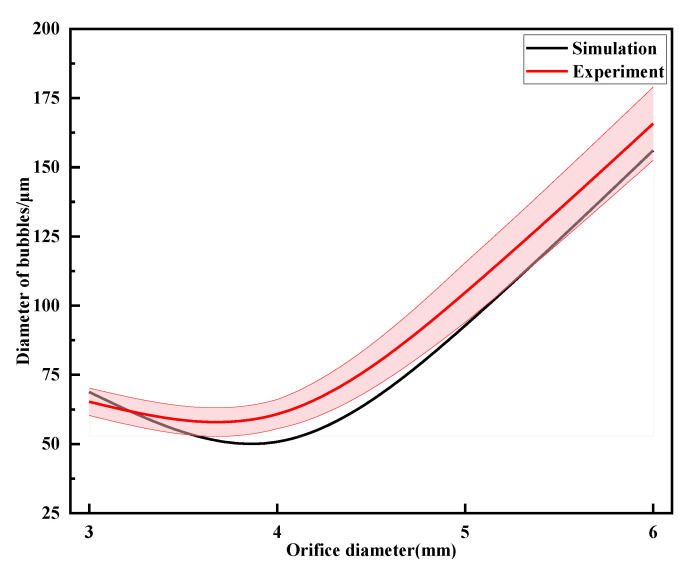
Results of changes in bubble particle size under different throttling apertures.

**Figure 21 micromachines-15-01025-f021:**
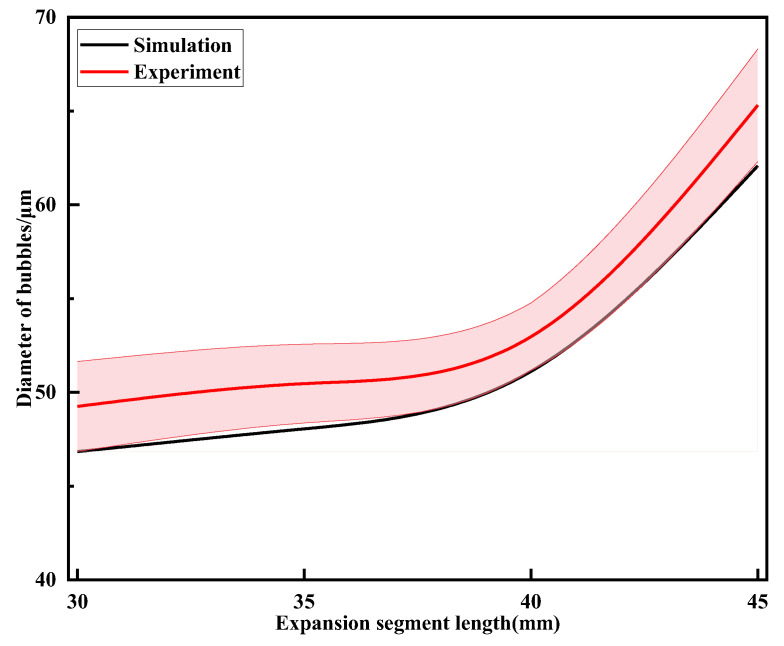
Results of changes in bubble size under different expansion section lengths.

**Table 1 micromachines-15-01025-t001:** Mesh Independence Verification Results.

Number	Grid Number	Average Turbulent Flow Dissipation Rate/ $m^{2}\cdot s^{-3}$	Relative Deviation/%
1	85,427	110.98	-
2	114,362	105.73	4.73
3	206,523	104.85	0.83
4	334,562	104.45	0.38

**Table 2 micromachines-15-01025-t002:** Results of Physical Model Testing and Validation.

Model	Pressure Drop across the Inlet and Outlet/kPa		Error Value/%
	Simulated Value	Experimental Value	
Standard k−ε	8.84	10.53	16.05
RNG k−ε	10.41	10.53	1.15
Reynolds stress	13.10	10.53	24.41

**Table 3 micromachines-15-01025-t003:** Parameter settings in the model.

Throttling Orifice Diameter r (mm)	Expansion Section Length D (mm)	Inlet Velocity v (m·s^−1^)
3	30	1
4	35	2
5	40	3
6	45	4

## Data Availability

The data are contained within the article.

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
