# Peer review of "Fragmentation Characteristics of Bubbles in a Throttling Hole Pipe"

_micromachines, 2024, doi:10.3390/mi15081025_

Round 1

Reviewer 1 Report

Comments and Suggestions for Authors

This manuscript aims to investigate the fragmentation characteristics of bubbles in a throttling hole adopting experimental and computational methods. The effect of the throttling hole pipe inlet speed, throttling hole diameter, and expansion section length on bubble fragmentation performance were analyzed. The numerical results are agreement with the experiments. It may provide some information about microbubble generators. However, the novelty of this work should be enhanced in the Introduction, where many other methods generate the microbubble should be included here.

Levitsky, I., Tavor, D. & Erenburg, V. A new bubble generator for creation of large quantity of bubbles with controlled diameters. Exp. Comput. Multiph. Flow 4, 45–51 (2022). https://doi.org/10.1007/s42757-020-0085-z.

Liao, Y., Ma, T. Study on bubble-induced turbulence in pipes and containers with Reynolds-stress models. Exp. Comput. Multiph. Flow 4, 121–132 (2022). https://doi.org/10.1007/s42757-021-0128-0

Mahmoudi, S., Saeedipour, M. & Hlawitschka, M.W. Bubble dynamics under the influence of the Marangoni force induced by a stratified field of contamination. Exp. Comput. Multiph. Flow (2024). https://doi.org/10.1007/s42757-023-0182-x

Bingbing Wang et al. Preparation and Properties of CO2 Micro-NanobubbleWater Based on Response Surface Methodology. Applied Surface.

Keiji Yasuda, Hodaka Matsushima, Yoshiyuki Asakura. Generation and reduction of bulk nanobubbles by ultrasonic irradiation. Chemical Engineering Science 195 (2019) 455–461

Mian Wu, et al. Generation of micro-nano bubbles by self-developed swirl-type micro-nano bubble generator. Chemical Engineering & Processing: Process Intensification 181 (2022) 109136.

We should the author should refer these works. We suggest this work should be tremendously revised.

1. The significance of this present work should be strongly enhanced by the reviewing the recent works. Some related works should be included in present reference. The introduction should be enhanced according to the excellent papers you can search in the ScienceDirect or others.

2.The number of bubbles can be identified using MATLAB. We may have a doubt about this because the image only can be captured in one plane. How do you deal with them in three dimensions because the experimental tube is three dimensional. Please clarify them.

3.The numerical simulation were done in present work using software COMSOL. Where are the governing Equations for multiphase flow? For example, mass equation and N-S equation. It is a large problem. If the governing equations are not given, your work is only a technical report.

4 The aim of this work is to study the fragmentation characteristics of bubbles. Please describe the model of bubble fragmentation? It can directly affect the novelty of your work. I think there are many similar works focused on the bubble fragmentation. You also have a comparison with previous works and present your novelty or views.

5. Please add the error bar in the graph including experimental results.

Comments on the Quality of English Language

none

Author Response

Dear Reviewer,

Thank you very much for your constructive comments. I have revised the manuscript according to your suggestions. Please see the attached file “reply1” for details.

Wishing you a wonderful day!

Best regards,  

Reviewer 2 Report

Comments and Suggestions for Authors

I would like to thank the author for conducting a neat work on tubular microbubble generator which will be helpful for future researchers and industries which are working in this domain. This work provides a clear idea of the critical parameters of a throttle-type microbubble generator and how it can be optimized for superior performance.

The good point about the paper is that it clearly states the objective of the paper and gives a detailed description of the experimental setup and structural diagram from which future researchers can recreate and extend the work. The choice of words were very simple and effective. They have conducted the necessary steps for a successful numerical study like grid independence study and case validation against test data.

There are some concerns that should be noted-

1.       The can be seen a significant deviation of Experimental and Computational results in Figure 18 and Figure 20 where COMSOL is under-predicting bubble size even though the trendline is well aligned with the test result. I would suggest the authors to re-run validation cases adjusting the parameter or mesh sizes. For future work they can also use a different CFD solver which will bring more confidence in their result.

2.       Section 5 (Line 419) and Section 6 (Line 465) have similar names, Section 6 supposed to be titled as “Conclusion” or something alike.

3.       Fig 3, Fig 7 is not clear, can be zoomed more.

Overall the work has some important findings that opens scope for future studies. If the authors fix the problems it would be an excellent addition to the academia.

Author Response

Dear Reviewer,

Thank you very much for your constructive comments. I have revised the manuscript according to your suggestions. Please see the attached file “reply2” for details.

Wishing you a wonderful day!

Best regards,  
